# The Importance of Phosphate Control in Chronic Kidney Disease

**DOI:** 10.3390/nu13051670

**Published:** 2021-05-14

**Authors:** Ken Tsuchiya, Taro Akihisa

**Affiliations:** 1Department of Blood Purification, Tokyo Women’s Medical University, Tokyo 162-8666, Japan; 2Department of Nephrology, Tokyo Women’s Medical University, Tokyo 162-8666, Japan; taro09071031@gmail.com

**Keywords:** CKD-MBD, FGF23, aKlotho, phosphate-binder

## Abstract

A series of problems including osteopathy, abnormal serum data, and vascular calcification associated with chronic kidney disease (CKD) are now collectively called CKD-mineral bone disease (CKD-MBD). The pathophysiology of CKD-MBD is becoming clear with the emerging of αKlotho, originally identified as a progeria-causing protein, and bone-derived phosphaturic fibroblast growth factor 23 (FGF23) as associated factors. Meanwhile, compared with calcium and parathyroid hormone, which have long been linked with CKD-MBD, phosphate is now attracting more attention because of its association with complications and outcomes. Incidentally, as the pivotal roles of FGF23 and αKlotho in phosphate metabolism have been unveiled, how phosphate metabolism and hyperphosphatemia are involved in CKD-MBD and how they can be clinically treated have become of great interest. Thus, the aim of this review is reconsider CKD-MBD from the viewpoint of phosphorus, its involvement in the pathophysiology, causing complications, therapeutic approach based on the clinical evidence, and clarifying the importance of phosphorus management.

## 1. Introduction

The series of changes that occur in renal dysfunction, including decreases in serum calcium levels due to impairment in vitamin D activation, hypersecretion of parathyroid hormone (PTH; i.e., secondary hyperparathyroidism), bone decalcification, and weak bones (osteomalacia), were collectively regarded as renal osteodystrophy. Following the introduction of the term “chronic kidney disease” (CKD), which expands the prior concept of renal dysfunction or renal failure, CKD-associated metabolic disorders of minerals (e.g., calcium) are now defined as CKD-mineral bone disease (CKD-MBD). As if coinciding with these changes, αKlotho was first reported in 1997, a deficiency of which results in an aging phenotype [1], and fibroblast growth factor 23 (FGF23), which causes a form of rickets and hypophosphatemia in bone metastasis, was identified in the early 2000s [2,3]. Phenotypical similarities between αKlotho-deficient mice and FGF23-deficient mice indicated an association between these two molecules, and their considerable involvement in the regulation of phosphate metabolism was subsequently revealed [4]. Given the significance of phosphate metabolism in CKD, and that αKlotho and FGF23 are involved in the pathophysiology as well as the onset of complications and survival in CKD, the importance of the αKlotho/FGF23 regulatory system in CKD-MBD is becoming clearer. This article reviews phosphate metabolism in CKD-MBD and its clinical significance, with a particular focus on the role of αKlotho/FGF23.

## 2. Basics of Phosphate Metabolism

Phosphate, along with calcium, is abundant in bone and as a component of hydroxyapatite, is essential for bone formation. It also has a variety of roles in cell biology (both in cell functions and maintaining life), including being a component of cell membranes and nucleic acids, being a component of ATP (the cell’s energy source), and regulating intracellular signaling via phosphorylation to control the function of enzymes and adjust pH. Phosphate ions are the most abundant intracellular anions, and its intracellular concentration is higher than that of serum. The metabolic balance of phosphate is regulated mainly by maintenance of the phosphate pool in bone and soft tissues, through phosphate absorption in the intestinal tract, osteogenesis, bone resorption, and excretion and reabsorption in the kidneys. Vitamin D and PTH have long been known to play major roles in the regulation [5]. The largest quantity of phosphate is stored in the bones, but the mechanism by which the phosphate pool is maintained in the bones has not yet been fully elucidated, and this is why many aspects of phosphate storage and its control in renal dysfunction or during dialysis remain unknown. Recently, sodium–phosphate cotransporters have been explored, and their critical role in phosphate reabsorption and excretion in the kidney was demonstrated [6]. The movement of phosphate in and out of the body basically relies on phosphate transporters. Sodium-dependent phosphate transporters (Na/Pi) that localize on proximal tubular epithelial cells and small intestinal epithelial cells are important for phosphate homeostasis in blood because they are responsible for excretion and reabsorption of phosphate in the kidney as well as phosphate absorption in the intestinal tract [7]. Approximately 80% of filtered phosphate in urine is reabsorbed in the kidney; of this, 60% is reabsorbed by the proximal convoluted tubules, 10 to 20% by the proximal straight tubule, whereas less than 10% is reabsorbed by the distal convoluted tubules.

Phosphate uptake can be roughly classified into intercellular transport and Na/Pi-mediated cellular transport. Na/Pi are proteins essential for phosphate uptake into cells via the plasma membrane. Three Na/Pi families have been identified. Type I includes gene, SLC17 A1 (Na/Pi-Ⅰ), and Type II includes the gene, SLC34 family (Na/Pi-Ⅱ). Both types are expressed in the kidney and small intestine and are responsible for epithelial phosphate transport. Na/Pi-Ⅱ is further classified into Ⅱa, Ⅱb, and Ⅱc. In proximal tubular epithelial cells, Na/Pi-Ⅱa and Na/Pi-Ⅱc are localized on the brush border membrane and play a pivotal role in phosphate reabsorption. Type III includes the gene, SLC20 family (PiT1, PiT2), which are expressed in various organs and are responsible for high-affinity phosphate transport. With advances in the cloning of genes that encode these sodium-dependent phosphate transporters and the elucidation of phosphate transport kinetics and related molecular mechanisms (e.g., molecular structures), regulatory factors such as FGF23 and associated small molecules have been identified and their functions have been clarified. These molecules are becoming the targets of new drugs [8]. It was long thought that PTH (a hormone that regulates calcium and phosphate levels) and 1,25-dihydroxyvitamin D (1,25(OH)2D) control Na/Pi to maintain the phosphate balance. Furthermore, the FGF23-mediated regulatory mechanism secreted from bone cells, which has recently emerged, is also affecting Na/Pi transport activity, that connects the kidney and bone in which αKlotho is expressed. Taken together, it is clear that many organs are involved in the maintenance of phosphorus homeostasis.

## 3. The αKlotho/FGF23 Axis

Since the phosphate-related molecules αKlotho and FGF23 were identified in the 1990s, powerful phosphate regulatory mechanisms have been revealed. In patients with abnormal laboratory test results indicative of hyperphosphatemia or secondary hyperparathyroidism, soft tissue calcification (e.g., vascular calcification) can occur in addition to renal osteopathy, and this can lead to fracture, cardiovascular event, and death. This systemic condition is now understood as a disorder of mineral and bone metabolism resulting from CKD, and the term CKD-MBD was proposed [9]. αKlotho is a factor that supports this disease concept. The aklotho gene attracted attention for its characteristic aging phenotype, consisting of a short life span, arterial calcification, pulmonary emphysema, and osteoporosis, in mice carrying an insertion mutation. Furthermore, the high mRNA expression level in the kidney and the development of hyperphosphatemia in insertion mutants suggested the possible involvement in the pathophysiology of kidney diseases [1]. αKlotho protein is composed of domains that show similarities to a carbohydrate-degrading enzyme (i.e., β-glucosidase), and exists in 2 forms (membrane-bound or secreted). The membrane-bound form is involved in the signal transduction of phosphaturic FGF23 by serving as a cofactor of the FGF23 receptor. More precisely, membrane-bound αKlotho acts as a co-replicator of the FGF23 receptor to enhance the specificity of the receptor and FGF23 actions [10,11]. FGF receptors are tyrosine kinase receptors encoded by 4 genes and make a complex with αKlotho to form the high-affinity FGF23 binding site [12]. However, αKlotho was originally recognized as a protein with multipotency, and various mechanisms of action were speculated, including the action of secreted αKlotho. Meanwhile, it was recently reported that the structure of αKlotho was not compatible with its glycosidase activity, suggesting that shed αKlotho functions as an on-demand non-enzymatic scaffold to promote FGF23 signaling [13].

The role of secreted aKlotho remains largely unknown. Secreted aKlotho has sialidase activity and cleaves sialic acids from N-linked glycans of glycoproteins, thereby contributing to the stability of glycoproteins in the membrane [14]. One possible role is preventing endocytosis of cation channels such as TRPV5 and ROMK1, thereby stabilizing them and establishing their calcium channel functions [15,16]. The regulation of TRPC6 channel by secreted αKlotho is reported to be a direct channel-gating action, independent of FGF23 receptors [17,18]. The inhibition of sodium-dependent phosphate transporters (Na/Pi-II), which are located in proximal tubules, inhibits the reabsorption of phosphate in the kidney, resulting in phosphate diuresis. However, given that αKlotho is localized mainly in the distal tubules [19], whereas Na/Pi-II is localized in the proximal tubules, it is possible that FGF23 interacts with the αKlotho–FGF complex to exert a paracrine action on adjacent proximal tubules; alternatively, the secreted form of αKlotho, which has glucuronidase activity, modifies Na/Pi-II glycans to decrease enzymatic activity or enhance the internalization of transporters, thereby modulating the inhibitory regulation [20]. Despite the several reported functions described above, many aspects of the mechanism by which αKlotho and FGF23 regulate Na/Pi-II remain to be elucidated [11], particularly, the mode of action of the secreted form of αKlotho and how it works in distant organs.

FGF23, a member of the fibroblast growth factor family, was identified as a phosphate diuretic factor. Around the same time, the FGF23 gene was identified as a causative gene of autosomal dominant hypophosphatemic rickets [2] and as a humoral factor in tumor-induced osteomalacia [3]. FGF23 is a 26-kDa protein comprising 251 amino acids and is produced and secreted mainly by osteocytes [21]. Phenotypic similarities between FGF23-knockout mice and αKlotho-knockout mice indicated an association between FGF23 and αKlotho, which were shown to be components of the same signal transduction system [4]. FGF family members (22 humoral factors) interact with FGF receptors and are involved in embryogenesis and organ development. Although FGF23 has a low affinity for FGF, the formation of a complex between αKlotho and FGF receptor 1c, 3c, or 4 creates a high-affinity binding site that exert actions in the kidney. The basic biological and physiological actions of FGF23 in calcium/phosphate metabolism is becoming clear. Shimada et al. administered synthetic FGF23 to experimental animals that had undergone parathyroidectomy and examined the effect of FGF23 without PTH action. The administration of synthetic FGF23 decreased serum phosphate levels, likely due to decreased phosphate reabsorption, which was indicated by the suppressed expression of Na/Pi-II in the proximal tubules. In addition, FGF23 decreased the expression of 1α-hydroxylase mRNA and increased the expression of 25-hydroxyvitamin D-24 hydroxylase mRNA, indicating that FGF23 suppresses the active form of vitamin D [22].

## 4. Phosphate-αKlotho/FGF23 Axis

It has been reported that the renal expression of αKlotho is reduced in CKD [23]. Sakan et al. examined biopsy specimens and showed progressive reduction in αKlotho immunostaining and in αKlotho mRNA expression from the early stage of CKD [24]. The expression of αKlotho is known to be decreased due to various types of stress and ischemia. For example, experiments using αKlotho-expressing cultured cells showed that oxidative stress decreased αKlotho expression [25]. These results indicated that decreased αKlotho expression not only causes metabolic disorders but may also elicit distinctive pathologies. αKlotho is reported to inhibit aging, increase lifespan, and prevent tissue damage, and thus it is easy to imagine an association between decreased expression of αKlotho and progression of CKD [26,27,28]. An experimental CKD model using α*klotho* (*+/−*) mice with ureteral obstruction showed aggravation of interstitial fibrosis in mice with reduced αKlotho expression, probably due to impaired suppression of TGFβ [29], suggesting that decreases in αKlotho expression weaken the mechanism that suppresses fibrosis. This further suggests a vicious cycle in which stress and ischemia lead to decreased expression of αKlotho in the renal tissue, which accelerates tissue damage during the progression of CKD (Figure 1).

αKlotho levels in serum and urine have been reported in many studies. The secreted form of αKlotho can be measured, but the correlations of serum αKlotho level with the degree of renal dysfunction and serum phosphate level remain controversial. Decreased urinary excretion of αKlotho and severe renal impairment have been reported in acute kidney injury [30], and its negative correlation with renal function has also been reported [31]. However, Seiler et al. showed that serum αKlotho was correlated with age but not with glomerular filtration rate (GFR), serum calcium level, or serum phosphate level and, compared with FGF23, it was not strongly associated with outcomes in CKD [32]. Whether αKlotho level can be a predictor of outcomes in CKD has not been well studied. In dialysis patients, rates of coronary artery diseases and left ventricular dysfunction were high when the serum αKlotho level was low, but a low serum αKlotho level was not necessarily correlated with cardiovascular disease or aortic calcification score [33]. Also, the association of secreted αKlotho with renal function and prognosis in CKD patients was questioned in a meta-analysis [34]. The measurement system is an important component in the testing of clinical specimens, but there are several limitations in αKlotho measurement that need to be addressed before the significance of αKlotho can be determined [35]. In addition, Ref. [36] summarized important measurement points and soluble αKlotho measurement methods to date. After all, establishment of new reproducible measurement methods, sample management, etc. are indispensable for clarifying the significance of human soluble αKlotho and its relationship with prognosis and pathophysiology. Taken together, it seems certain that αKlotho expression decreases in the renal parenchyma, but many issues still need to be addressed including analysis of the secreted form of αKlotho; investigation of the mode of action of the secreted form of αKlotho and its actions in distant organs; measurement of serum αKlotho level in renal failure with decreased renal αKlotho expression; and development of reproducible measurement techniques. Nevertheless, αKlotho has many potential implications for diagnosis, physiological action, and pathophysiology [36,37]. 

However, in pre-dialysis CKD, increases in serum phosphate level occur at later stages and therefore hyperphosphatemia is not always detected early. As renal dysfunction progresses, renal αKlotho expression decreases while FGF23 and PTH increase, leading to increased phosphate excretion per nephron to compensate for the decreased GFR [38,39] (Figure 2). It should be noted that FGF23 increases in the early stage of CKD, but at present it is unclear whether αKlotho decreases [40]. FGF23 is likely to have a suppressive effect on phosphate storage attributed to clearance impairment, but early increases in FGF23 level, even when the serum phosphate level is in the normal range, could suggest the presence of other stimulatory factors. A prospective and cohort study involving 3879 patients with a mean eGFR of 42.8 mL/min showed that an increased FGF23 level predicted the prognosis of CKD, specifically, a significantly high mortality and disease progression to end-stage kidney disease [41]. Prognosis was poor in the group with hyperphosphatemia at the initiation of dialysis, and an increased FGF23 level was a risk factor of mortality [42]. FGF23 was also reported to be a predictor of cardiovascular events in predialysis CKD patients but did not predict events at the initiation of dialysis and during maintenance dialysis [43]. The significance of these reported high levels of FGF23 during maintenance dialysis remains inconclusive.

αKlotho and FGF23 have been reported to exert physiological activities and direct biological actions, respectively. Also, the multipotency of αKlotho is of great interest. Transgenic mice overexpressing αKlotho protein are reported to have a 20–30% longer lifespan, possibly through a mechanism involving the suppression of insulin/insulin-like growth factor 1 (IGF1) signaling by αKlotho [26]. Generally, intracellular insulin/IGF1 signaling is involved in the activation of redox signaling, the cytotoxicity of which induces aging. It has also been suggested that αKlotho influences Wnt signaling [27] and induces enzymes that scavenge reactive oxygen species to regulate apoptosis, thereby protecting proteins and DNA from damage by oxidative stress as well as suppressing tissue damage [28]. Meanwhile, several studies have reported the activities of FGF23. FGF23 was reported to induce cardiac hypertrophy in an animal model and to enhance sodium reabsorption in humans, thereby inducing hypertension [44,45]. These reports are in good agreement with the association between FGF23 and the risk of chronic heart failure [46]. However, a meta-analysis compared participants in the top third with those in the bottom third of FGF23 concentration and showed that the summary relative ratios (95% confidence intervals [CI]) were 1.33 (1.12–1.58) for myocardial infarction, 1.26 (1.13–1.41) for stroke, 1.48 (1.29–1.69) for heart failure, 1.42 (1.27–1.60) for cardiovascular mortality, and 1.70 (1.52–1.91) for all-cause mortality. For non-cardiovascular mortality, the summary relative ratio was 1.52 (95% CI, 1.28–1.79). These results suggest no causal relationship between FGF23 and cardiovascular disease risk [47]. In the above studies, the FGF level in CKD was measured at a single point in time, rather than over a given period. A prospective and case-cohort study involving chronic renal insufficiency patients monitored the FGF23 level at 2–5 years (mean, 4.0 ± 1.2 years) in a randomly selected sub-cohort of 1135 participants. The median FGF23 level was stable for 5 years of follow-up, but the distribution gradually skewed to the right, suggesting a subpopulation with a markedly elevated FGF23 level. Compared with participants with stable FGF23 levels, those with slowly increasing FGF23 levels had a 4.49-fold higher risk of death (95% CI, 3.17–6.35), whereas those with rapidly increasing FGF23 levels had a 15.23-fold higher risk of death (95% CI, 8.24–28.14). Conclusively, FGF23 levels are stable over time in the majority of CKD patients, but monitoring can identify subpopulations with elevated FGF23 levels and an extremely high risk of death [48]. The biological activities and clinical significance of soluble αKlotho and FGF23 remain largely unknown and further studies are anticipated [49,50].

## 5. Phosphate Metabolism in CKD

As described earlier, CKD-MBD develops in the early stage of CKD and persists in the dialysis stage after renal function is abolished. The pathophysiology of CKD-MBD has traditionally included vitamin D activation disorder and hypocalcemia due to renal dysfunction; hyperparathyroidism secondary to those stimuli; and bone changes due to elevated PTH levels. Phosphate metabolism disorders and complications such as vascular lesions are also attracting attention. In pre-dialysis CKD, the serum phosphate level starts rising at the late stage, and thus hyperphosphatemia is unlikely to be detected at the early stage. Decreases in GFR lead to increases in phosphate excretion per nephron, as well as increases in the serum level of phosphaturic factors (i.e., PTH and FGF23). These changes are likely caused by the decreased expression of αKlotho in the kidney. However, it remains unclear how the membrane-bound form and the soluble form of αKlotho are involved. Compensatory increases in PTH and FGF23 when serum phosphate levels are within the normal range are reported to be predictors of high mortality [39]. 

In the dialysis stage, hyperphosphatemia also influences outcomes, for example, by causing vascular calcification. Another cause of hyperphosphatemia is secondary hyperparathyroidism. The serum phosphate level increases due to PTH-dependent bone resorption. In addition, ectopic calcification progresses in the early stage of CKD, suggesting possible phosphate accumulation in soft tissue as well as in bone. When renal function is impaired, excessive phosphate increases PTH secretion and decreases the active form of vitamin D, which in turn induces secondary hyperparathyroidism.

## 6. Outcomes and Complications in Hyperphosphatemia

### 6.1. Effect on Renal Function Impairment and on Survival

Generally, in pre-dialysis CKD, the serum phosphate level remains within the normal range until reaching stage 4 or 5. Therefore, it is unclear what kind of prognostic role the serum phosphate level plays. Meanwhile, the significance of the serum phosphate level in a population without kidney disease was reported in a study examining the relationship between phosphate level and the onset of renal failure. Although the phosphate level was thought to be unrelated to the onset and progression of CKD in healthy individuals, a retrospective longitudinal cohort study was performed in the period 1 January 1998 through 31 December 2008 of adults within a vertically integrated health plana. A total of 94,989 individuals within the health plan showed that the risk of end-stage renal disease was significantly higher (the hazard ratio, 1.48) in patients in the 4th (highest) phosphate quartile compared with those in the 1st (lowest) phosphate quartile, suggesting that a relatively high phosphate level within the normal range can be a risk factor for CKD in healthy individuals [51].

Several studies have investigated the relationship of serum phosphate level with renal prognosis and mortality. In post hoc analysis of outcome data retrieved from the cohort of 331 patients with chronic proteinuric nephropathies included in the REIN trial, which is a study examined the progression of renal dysfunction and responses to angiotensin-converting enzyme inhibitor, the proportion of patients who progressed to end-stage renal diseases and a group of faster serum creatinine doubling time was significantly higher in patients with serum phosphate levels in higher quartiles. Also, responsiveness to ramipril, a renin-angiotensin converting inhibitor, was decreased, suggesting the possible influence of serum phosphate level on responsiveness to drugs [52]. A prospective cohort study involving 6730 patients with CKD (defined by elevated serum creatinine level) showed that among the 3490 patients in whom phosphate measurements were performed, a serum phosphate level ≥3.5 mg/dL was associated with significantly increased mortality risk, which increased linearly with each subsequent 0.5 mg/dL increase in serum phosphate level. In addition, the mortality risk nearly doubled in patients with a serum phosphate level ≥4.5 mg/dL compared with those with a normal serum phosphate level (2.5–2.99 mg/dL) [53]. Furthermore, when the phosphate level was already high in patients with stage 4–5 CKD, the impairment of renal function progressed more rapidly, and the crude mortality risk was increased to 1.62 for every 1-mg/dL increase in phosphate level [54]. A meta-analysis of 12 cohort studies involving a total of 25,546 patients showed that 8.8% developed kidney failure and 13.6% died, and every 1-mg/dL increase in serum phosphate level was associated with kidney failure (hazard ratio, 1.36) and mortality (hazard ratio, 1.20) [55].

Since the 1990s, phosphate has been shown to be a clear risk factor in the dialysis stage, and this is mentioned in several clinical practice and treatment guidelines [56,57,58]. A 2011 meta-analysis reported that the mortality risk increased by 18% for every 1-mg/dL increase in the serum phosphate level (relative risk [RR], 1.18; 95% CI, 1.12–1.25). There were no significant association of all-cause mortality with serum PTH level (RR per 100-pg/mL increase, 1.01; 95% CI, 1.00–1.02) or serum calcium level (RR per 1-mg/mL increase, 1.01; 95% CI, 1.00–1.162), indicating the importance of phosphate compared with PTH and calcium [58]. Although the degrees of importance were compared only in a limited number of studies [59], the monitoring of phosphate, calcium, and PTH levels in a 3-year cohort of 128,125 hemodialysis patients in Japan showed that mortality was lowest in patients who achieved the phosphate target compared with those who achieved the targets of other markers, indicating that the phosphate level was the strongest predictor of mortality, followed by calcium and PTH [60].

### 6.2. Hyperphosphatemia and Vascular Calcification

Vascular calcification has been regarded as an aging-associated phenomenon, and conventional risk factors include aging, diabetes, lipid abnormality, and hypertension. In addition, an association with CKD-MBD was recently revealed. Vascular calcification is an important component in CKD-MBD and influences patients’ survival [9,61].

Calcification occurs at the vascular intima and the vascular media, and intimal lesions are associated with atherosclerosis. Intimal lesions are characterized by lipid deposition and infiltration by inflammatory cells such as macrophages, resulting in the formation of protruding lesions called plaques. In dialysis patients, these plaques are often associated with a high degree of calcification. On the other hand, medial calcification is known as Mönchsberg’s calcification, which is pathologically characterized by calcium deposition within the medial tissues of muscular arteries, sometimes with osseous metaplasia. Medial calcification is reported to be associated with all-cause and cardiovascular mortality and induces arteriosclerosis [62]. The molecular mechanism underlying vascular calcification is becoming clear. In addition to traditional factors such as phosphate, the involvement of new factors like αKlotho and FGF23 as well as regulatory factors at the cellular level, have been identified [63]. Given that proteins associated with bone (e.g., osteopontin, osteocalcin, osteoprotegerin, and matrix Gla protein), which are produced by osteoblasts and chondrocytes, are present in calcified lesions, the transformation of vascular smooth muscle cells is likely to play a central role [64]. Accordingly, the basic mechanism, which involves cellular phosphate uptake via the phosphate transporter Pit-1, followed by transformation of vascular smooth cells into osteoblasts and chondrocytes and then induction of medial calcification, indicated that phosphate is a strong factor in calcification [65]. Additional details of the pathophysiology have been reported, including the enhancement of osteochondrogenic differentiation, the induction of apoptosis, and the fibrosis and mineralization of extracellular matrix [66]. As the association between vascular calcification and phosphate became clearer, several related factors were reported. The Chronic Renal Insufficiency Cohort (CRIC) study revealed that serum phosphate was associated with the coronary artery calcification score in pre-dialysis CKD patients, but that FGF23, which influences the cardiovascular system, had no association with and no influence on coronary artery calcification [67]. In addition, the associations of serum phosphate concentrations with vascular and valvular calcification in 439 participants from the Multi-Ethnic Study of Atherosclerosis study who had moderate CKD and no clinical cardiovascular disease were examined, and high serum phosphate level within the normal range, after adjustment for PHT and vitamin D, was significantly associated with calcification of the coronary arteries and cardiac valves [68]. These studies suggest that the risk of vascular calcification exists from the pre-dialysis stage and that phosphate has a considerable influence on this risk, although it remains unclear what effect reducing the phosphate level might have.

Vascular calcification is very common in aging, diabetes and especially in CKD. Vascular calcification is a powerful predictor of cardiovascular morbidity and mortality in the CKD population. Elevated serum phosphate is a late symptom of CKD and has been shown to promote mineral deposition in both vascular walls and heart valves. αKlotho and FGF23 are new factors in CKD-MBD and are thought to be involved in the pathogenesis of uremic vascular calcification. There are inconsistent reports on the biomedical effects of FGF23, in contrast, increased evidence supports αKlotho’s protective role in vascular calcification.

### 6.3. Hyperphosphatemia and Fracture

The risks of osteoporosis and fracture are expected to increase with age in CKD patients [69]. In particular, the Dialysis Outcomes and Practice Patterns Study (DOPPS) report, which involved 34,579 dialysis patients in 12 countries, showed that the frequency of femoral neck fracture was higher in dialysis patients than in the general population of each participating country. Decreases in the quality of life and a high mortality rate were also reported. Osteoporosis and bone lesions due to renal dysfunction (traditionally called renal osteodystrophy and now called CKD-MBD) differ in terms of pathophysiology, but both are considered complications of aging [70]. A cohort study of 679,114 participants showed that the risk of fracture during the 3-year observation period was high in both sexes when eGFR was ≤15 mL/min and participants were aged ≥65 years [71]. A Japanese cohort study of 162,360 participants examined 5-year all-cause mortality and cause-specific mortality and showed that crude mortality rates doubled in participants with hip fracture; this higher mortality persisted during the 5-year period [72].

Several studies have reported a direct association between phosphate and fracture. Experiments using cultured osteoblast-like cells revealed that inorganic phosphate induces apoptosis [73]. In addition, increases in the phosphate concentration of cell culture media inhibited the RANK–RANKL signaling-mediated cell differentiation of cultured osteoclast-like cells [74]. Regarding the association between serum phosphate level and fracture in humans, an increased risk of fracture was reported in healthy individuals when the phosphate level was the mid- to upper normal range, and phosphate level was reported to be a significant risk factor for fracture in men with CKD [75,76].

## 7. Treatment

### Dietary Treatment of Hyperphosphatemia

Dietary treatment of CKD is based on the restriction of dietary protein. The Japanese clinical practice guidelines for CKD recommend restricting dietary protein to limit the progression of CKD (e.g., 0.8–1.0 g protein/kg standard bodyweight/day for stage G3a, and 0.6–0.8 g protein/kg standard bodyweight/day for stage G3b or later). Possible rationales include induction of glomerular hyperfiltration by excessive protein intake, which affects renal function, and the accumulation of uremic protein metabolites when renal function is impaired; however clinical studies have not produced conclusive evidence in support of these.

A meta-analysis of randomized controlled trials (RCTs) involving 779 diabetic kidney disease patients showed that GFR was 5.82 mL/min higher in patients on a protein-restricted diet (0.6–0.8 g protein/kg bodyweight/day) for 18 months compared with those who were not (1.0–1.6 g protein/kg bodyweight/day) [77]. Another meta-analysis showed that the estimated effect of a protein-restricted diet compared with the control diet on annual changes in GFR was −0.95 mL/min. The estimated effect for the non-diabetic and type 1 diabetic patients was −1.50 mL/min, whereas that for type 2 diabetic patients was −0.17 mL/min; the estimated effect for type 2 diabetic patients was not significant [78]. The results of a study that searched the Cochrane Kidney and Transplant Specialized Register up to 7 September 2020 were also reported. This study included 17 RCTs involving a total of 2996 non-diabetic pre-dialysis adult CKD patients with renal function impairment and showed that fewer patients progressed to the stage requiring dialysis when a very-low-protein diet (0.3–0.4 g protein/kg bodyweight/day) was provided than when an ordinary low-protein diet (0.5–0.6 g protein/kg bodyweight/day) or a normal-protein diet (0.8 g protein/kg bodyweight/day) was provided for 12 months. There was little difference in the number of patients with non-severe kidney failure who progressed to the stage requiring dialysis between the low-protein diets and normal diet. These results indicate that very low protein intake may delay the progression of kidney failure, but more information, including adverse effects, adherence difficulties, and the impact on quality of life, is needed [79].

Protein restriction has been a clinical proposition for a very long time, but the present situation is as described above. The difficulty of strictly adhering to the protocol, which makes comparisons inaccurate, may be the reason for the inconclusive effect. In addition, the concept of nutritional management changes with the age of the CKD patient. Possible impairments in physical and mental function (e.g., frailty) resulting from drastic protein restriction are of concern. Meanwhile, a study focusing on phosphorus as a harmful substance was also reported [80]. As described earlier, it should be noted that, in pre-dialysis CKD, the serum phosphate level is maintained for a long time because the phosphate excretion per nephron increases as the number of nephrons decreases. FGF23 from the bone is implicated in this, but high FGF23 levels are associated with poor prognosis. The question then arises as to whether restricting phosphorus intake in order to suppress FGF23 would be effective. However, this question remains unanswered. Also, as will be discussed later in this article, an attempt to use phosphate binders to reduce the level of FGF23 in patients with a normal serum phosphate level was unsuccessful.

Then, is protein restriction equal to phosphorus restriction? As the features of phosphorus, especially its role as a nutrient, have attracted increasing attention, the management of phosphorus, rather than protein, has become more important. A study (The NHANES is an ongoing series of the surveys of the non-institutionalized civilian population in the USA conducted by the National Center for Health Statistics. From 1988 to 1994, NHANES III, a cross-sectional survey of the US population was carried out.) examining the effect of dietary phosphorus intake in 1105 patients with stage 3 CKD (eGFR 49.3 ± 9.5 mL/min) showed that dietary phosphorus intake did not influence mortality when the serum phosphate level was in the normal range [81]. Also, analysis of the data from the Modification of Diet in Renal Disease (MDRD) study showed no association of 24-h urine phosphate excretion with progression to end-stage renal disease, all-cause mortality, and cause-specific mortality [82]. Meanwhile, a follow-up of 95 patients with stage 2–3 CKD for 2.7 ± 1.6 years showed a decline in eGFR of 0.5 mL/min/year and a correlation between the rate of eGFR decline and the degree of phosphaturia. In the same study, phosphate load caused renal toxicity in an animal model, suggesting that phosphorus directly damages renal tubules [83]. In addition, a study of patients on maintenance dialysis using the Food Frequency Questionnaire, which was developed to simultaneously obtain information about food intake habits and nutrient intake, showed that dietary phosphorus intake and the phosphorus-to-protein ratio were associated with mortality [84]. Taken together, the significance of phosphorus restriction in the pre-dialysis stage remains inconclusive.

The importance of phosphorus restriction has been reported in hyperphosphatemia, and the phosphorus-to-protein ratio and the characteristics of phosphorus itself, in addition to the absolute amount of dietary phosphorus, are considered important [85]. Dietary phosphorus can largely be classified as organic or inorganic, both of which have different absorption rates in the body. Organic phosphorus binds to proteins and phytic acid and is absorbed at a rate of about 50%. Organic phosphorus can be further classified by source, namely, plant or animal proteins. The organic phosphorus in animal-derived foods is readily digestible, so its absorption rate is relatively high, whereas that in plant-derived foods is often phytate-based, so its absorption is limited. Indeed, compared with animal-derived dietary protein, plant-derived dietary protein led to lower serum phosphate levels and FGF23 levels [86] indicating the usefulness of plant-derived dietary protein in the restriction of dietary phosphorus intake [87]. Meanwhile, inorganic phosphorus is a main component of food additives (e.g., acidifiers, emulsifiers, baking powder, and pH stabilizers) which are a recent cause of concern and are used in many processed foods.

As described earlier, the toxicity of phosphorus itself is considered a problem. The number of studies reporting an association of excessive phosphorus intake with CKD, cardiovascular disease, and bone lesions has been increasing, which likely reflects the excessive consumption of processed foods [88]. Given that calcium contents vary but phosphorus is present in all commonly eaten foods, food additives are likely to increase the phosphate load [89]. Given that dietary phosphorus intake has been increasing, and the harmful nature of phosphate especially in CKD and CKD-MBD has become a cause for concern, the use of phosphorus binders has become central to the management of dietary phosphorus intake. Several systematic reviews conducted in different years reported the effect of dietary educational interventions on the reduction of phosphate levels [90,91]. Dietary educational interventions had a suppressive effect on hyperphosphatemia, as shown in the latest review. Statistical significance was confirmed in some studies, but their reliability was questioned because of the randomization process and deviations from protocol. Monthly dietary educational interventions (20–30 min) reduced phosphate levels without compromising the nutrition status of patients with persistent hyperphosphatemia, but the effect did not persist when the interventions were discontinued. Nevertheless, trials varied widely in terms of design, approach, and so on, and thus the evidence obtained was very much limited [92].

The protein and phosphate contents as well as the phosphate-to-protein ratio are higher in processed foods than in fresh foods. Knowing the phosphate content of foods, especially processed foods, might contribute to better phosphate management in CKD patients with hyperphosphatemia [93]. Phosphorus-containing food additives are becoming a social problem. Given the high levels of such additives among the most popular foods sold in grocery stores as well as the low price of such foods, it is likely that CKD patients often purchase these products [94]. An RCT involving 279 dialysis patients with a high baseline serum phosphate level (≥5.5 mg/dL) examined the effect of dietary education. After 3 months, approximately 1.0-mg/dL or 0.4-mg/mL decreases in serum phosphate level were seen in those who received or did not receive the dietary educational intervention, respectively [95]. The effect of such educational programs on reducing phosphate levels was confirmed in another RCT study [96]. As pointed out, an administrative approach may be important for the management of dietary phosphate intake and, indeed, the restriction of dietary phosphate intake might be achieved through regulatory actions implemented by US Food and Drug Administration. Mandatory labeling of phosphate content on all packaged food and drugs would make it easier to quickly identify healthy low-phosphate food and drugs, which would in turn make it easier for individuals to control their total phosphate intake. Simple changes in regulatory policies and labeling are warranted to enable better management of dietary phosphate intake in all stages of kidney diseases as well as potentially reduce health risks in the general population [97].

## 8. Aging and Frailty

As described earlier in this article, a certain amount of protein intake is required for aging CKD patients to prevent malnutrition caused by protein-restricted diets. Protein-restricted diets may induce or aggravate impairment in physical function, sarcopenia, and frailty. Indeed, such problems and their outcomes in elderly patients with end-stage kidney disease have been reviewed [98]. Patients with frailty often have CKD. In such cases, the eGFR, which is used as a diagnosis criterion for CKD, should be carefully calculated because estimations based on creatinine level, which reflect the muscle mass, differ from those based on cystatin C level [99]. The prevalence of frailty is high in pre-dialysis CKD patients, and the presence of frailty was associated with the risk of progression to end-stage CKD [100]. A systematic review reported that the prevalence rates of frailty and impaired physical function were high in patients with CKD and that the rate of disease progression to end-stage CKD and mortality were ≥2-fold higher when frailty was present [101]. Similar trends were shown in a Japanese study [102]. The prevalence rates of conditions such as frailty and cognitive impairment were also high in elderly dialysis patients, and complications were seen in many patients [103,104]. The prevalence of frailty in 117 patients aged ≥69 years (mean age, 78.1 years) was high, and the HR for 12-month mortality was 2.6 in frail participants compared with non-frail participants [105]. Japanese clinical practice guidelines for CKD recommend appropriate management of CKD-MBD in elderly CKD patients aged ≥75 years. Restriction of dietary phosphate intake and administration of phosphate binders are recommended for the management of hyperphosphatemia, and care should be taken to avoid decline in appetite and nutritional status, especially in elderly patients. A cohort study (historical cohort) including pre-dialysis CKD patients showed an approximately 40% lower risk of all-cause mortality in those who received oral phosphate binders compared with those who did not. This association was also seen in a subgroup aged ≥71 years, suggesting that administration of phosphate binders might improve outcomes in elderly patients [106]. A cohort study of dialysis patients also showed an association between serum phosphate level and risk of mortality in elderly patients. In this study, 107,817 hemodialysis patients in the US were followed for 6 years and increases in all-cause mortality risk were associated with elevated serum phosphate level in subgroups of those aged 70–74 years, ≥75 years, and in the younger subgroup [107]. These results indicate that, even when a certain degree of protein intake is ensured in elderly patients, phosphate management should not be neglected; rather, adequate management is required. A RCT of hemodialysis patients demonstrated the superiority of the phosphate-binding medication sevelamer to calcium-based phosphate binders in lowering the mortality only in the subgroups aged ≥65 years, indicating that calcium loading should also be avoided in elderly patients [108].

## 9. The Significance of Phosphate Binders

Ensuring a certain amount of calories and protein intake and protecting against increases in phosphate level are likely to affect the prognosis of CKD, especially in elderly patients (as described earlier). Protein restriction, which is central to the dietary treatment of CKD, may lead to the impairment of physical function, including frailty, especially in elderly patients. Thus, the recent trend in therapy is to use two conflicting strategies, namely, ensuring sufficient protein intake and restricting phosphate intake. If dietary therapy, as described earlier, is not satisfactory, then the use of phosphate binders needs to be considered.

### 9.1. Administration of Phosphate Binders in Pre-Dialysis and Dialysis Stages

When the serum phosphate level does not increase in pre-dialysis CKD patients, it is likely that the pathophysiology of CKD-MBD described earlier, which includes increases in FGF23 level and downregulation of αKlotho (although it is difficult to prove clinically), has already begun. If phosphate binders (e.g., sevelamer and lanthanum carbonate) can reduce the FGF23 level by decreasing the phosphate load, parameters indicative of poor outcomes may improve. In a study involving CKD patients without hyperphosphatemia (eGFR, 20–45 mL/min/1.73 m^2^) (RCT study) who were randomly assigned to receive placebo (58 patients), lanthanum carbonate (30 patients), sevelamer carbonate (30 patients), or calcium acetate (30 patients), those who received phosphate binders showed significant decreases in serum phosphate level (from 4.2 mg/dL to 3.9 mg/dL on average) and urinary phosphate level. In addition, the attenuation of secondary hyperparathyroidism was significantly higher in the groups receiving phosphate binders compared with the placebo group, but phosphate binders did not affect the FGF23 level. Taken together, the effect of phosphate binders on FGF23 levels cannot be expected in pre-dialysis CKD patients without hyperphosphatemia, and thus a dietary therapy-based strategy was recommended [109]. Meanwhile, an observational study demonstrated the usefulness of phosphate binders in CKD complicated by hyperphosphatemia. This study examined the association of using an oral phosphate binder (sevelamer or a calcium-based phosphate binder) with mortality and the eGFR slope in 1188 veterans with CKD. The mean eGFR was 38 ± 17 mL/min/1.73 m^2^, and the majority of participants had stage 3 (57%) or stage 4 (30%) CKD. The analysis revealed an association between the use of oral phosphate binders and low mortality (adjusted HR, 0.61; 95% CI, 0.45–0.81) [106].

In dialysis patients, the administration of phosphate binders led to favorable outcomes. The DOPPS, which was a prospective cohort study involving 23,898 dialysis patients from 12 countries, showed that 88% of patients were prescribed phosphate binders and that the mortality rate was 25% lower in those patients than in those without prescription, but only when the phosphate level was ≥3.5 mg/dL [110]. In the COSMOS trial (a 3-year follow-up, multicenter, open-cohort, observational prospective study) involving 6797 patients, phosphorus binding prescriptions also associated with the risk of total and cardiovascular mortality by 29% and 22%, respectively [111]. Intention-to-treat analyses (analyzed a prospective cohort study of 10,044 incident hemodialysis patients using Cox proportional hazards analyses to compare 1-year all-cause mortality among patients who were or were not treated with phosphate binders) also revealed the effect of phosphate binders on survival in dialysis patients [112]. Meanwhile, a meta-analysis of studies selected by a recent search of the Cochrane Kidney and Transplant Register of Studies showed that the use of sevelamer during dialysis was more effective in lowering the mortality compared with calcium-based phosphate binders, and that use of any of phosphate binders did not have a significant effect on myocardial infarction, stroke, fracture, or coronary artery calcification [113].

### 9.2. Types of Phosphate Binders and Their Characteristics

#### Calcium-Based Preparations

The state of using phosphate binders in pre-dialysis CKD was explained earlier in this article. Several studies have reported the effects of calcium-based preparations. Although the number of participants was limited, the comparison of calcium carbonate (1500 mg/day) and placebo showed that administration of calcium carbonate produced a positive calcium balance but did not affect the phosphate balance in patients with stage 3 or 4 CKD [114]. This means that administration of calcium carbonate has no influence on phosphate balance but does increase the calcium load. Meanwhile, a RCT of pre-dialysis CKD patients with hyperphosphatemia showed that administration of calcium acetate caused decreases in phosphate and PTH levels and increases in the calcium level [115].

Subsequently, calcium loading resulting from administration of phosphate binders became a problem, especially in Japan, where the use of these binders increased because aluminum-based binders were contraindicated for CKD patients. Calcium carbonate preparations were approved for the treatment of hyperphosphatemia, but given the clinical use of activated vitamin D (intravenous injection), problems arose, including hypercalcemia, excessive suppression of the parathyroid, adynamic bone disease, and calcification of soft tissues and blood vessels. For this reason, although calcium carbonate became the standard phosphate binder because of low cost, widespread use is to be avoided.

As a result, clinical studies comparing calcium-based agents and new resin-based alternatives were conducted. Sevelamer is a cationic polymeric resin that binds to free phosphate derived from food within the gastrointestinal tract and is excreted in the stool without being degraded or absorbed, thereby suppressing phosphate absorption in the body. A RCT study of pre-dialysis patients with stage 3 or 4 CKD showed that all-cause mortality, the rate of dialysis inception, and composite endpoint (mortality plus dialysis inception) were significantly lower in the group that received sevelamer compared with the group that received calcium carbonate [116]. 

The negative influence of a calcium-based phosphate binder compared with a non-calcium-based phosphate binder on survival was also shown in a RCT study of dialysis patients [117]. A 2013 meta-analysis also showed a more favorable effect on survival by using a non-calcium-based phosphate binder compared with a calcium-based phosphate binder [118]. Although the superiority of resin-based phosphate binders to conventional calcium-based phosphate binders was reported [119], the results of a more recent meta-analysis were inconclusive [120]. Furthermore, there are emerging problems related to resin-based phosphate binders, including a high pill burden, non-adherence, and consequent poor control of serum phosphate level [121].

### 9.3. Metal-Based Phosphate Binders

#### 9.3.1. Lanthanum Carbonate Preparations

More recently, metal-based phosphate binders have been launched. Lanthanum carbonate was introduced in 2004, and its characteristics, which include better phosphate binding efficacies and lower pill burden compared with those of resin-based phosphate binders, are of great interest in the treatment of CKD, where polypharmacy is a considerable challenge. A study examining lanthanum carbonate in pre-dialysis patients (RCT) and a study comparing lanthanum carbonate with calcium carbonate in dialysis patients (a crossover study) showed that lanthanum carbonate successfully decreased the phosphate level and allowed for increased dose of vitamin D analogue without causing hypercalcemia [122,123]. A systematic review of RCTs and quasi-RCTs showed similar results, reporting that heavy metal accumulation in blood and bone was below toxic levels, albeit with a higher rate of vomiting compared with other agents [124]. Based on the experience with aluminum-based preparations, accumulation of heavy metals in the body is a cause of concern. However, paired bone biopsies confirmed no accumulation of lanthanum in patients treated with lanthanum carbonate over a long period of time [125]. On the other hand, whereas aluminum is excreted via renal pathway, lanthanum is predominantly via the hepatobiliary pathway [126]. The subcellular localization of lanthanum in liver tissue was determined using microscopy and spectroscopy techniques [127]. Lanthanum is present in the lysosomes of hepatocytes, and there are no reports of cells or tissue damage in the liver. Moreover, there was no evidence of an increase in the incidence or severity of liver-related adverse effects in patients who received lanthanum for up to 6 years [128].

Meanwhile, a meta-analysis of RCTs showed significantly lower mortality in the lanthanum carbonate-treated group compared with groups that received other agents, but no significant difference in the rate of cardiovascular events was observed between the groups [129]. Recently, lanthanum has attracted attention not only for its effect on phosphate levels but also on nutritional status. As mentioned earlier in this article, lowering the serum phosphate level without drastically restricting protein intake is considered beneficial in aging patients, and indeed, patients who received lanthanum carbonate showed improved mortality and nutritional status [130]. In addition, the proportion of hypoalbuminemic dialysis patients who achieved a ≥0.2 g/dL increase in serum albumin level and maintained their serum phosphate level within a range of 3.5–5.5 mg/dL was significantly higher when they consumed a high-protein diet containing 400–500 mg phosphorus with lanthanum carbonate compared with a low-protein diet containing ≤200 mg phosphorus with a conventional phosphate binder (27% vs. 12%) in RCT study [131]. Lanthanum carbonate has an inhibitory effect on calcium absorption [132] and so does not have a suppressive effect on PTH in pre-dialysis patients. In addition, lanthanum carbonate, in RCT of CKD stage 3b/4 patients without hyperphosphatemia, did not significantly affect on arterial stiffness, aortic calcification, and serum phosphorus, PTH, and FGF23 levels for 96 weeks [133].

#### 9.3.2. Ferric Citrate Preparations and Sucroferric Oxyhydroxide

Ferric citrate is a new type of phosphate binder, the active ingredient of which is ferric citrate hydrate. Ferric iron inhibits phosphorus absorption by binding to phosphorus in the gastrointestinal tract, thereby lowering the serum phosphate level. Its effective in lowering phosphorus is expected to be similar to that of other phosphate binders. It is classified as a metal-based agent, and in addition to its role as a phoshpate binder, it is expected to provide the beneficial activities of iron itself. 

Iron is expected to inhibit the osteochondrogenic transformation of vascular smooth muscle cells induced by high phosphate in cultured cells. It binds to phosphorus to inhibit phosphate transport and suppresses calcification by directly controlling transformation [134]. Furthermore, a RCT study involving 203 pre-dialysis CKD patients with eGFR ≤20 mL/min showed that compared with conventional phosphate binders, ferric iron significantly extended the time to hospitalization, death, dialysis inception, and transplantation. In addition, the FGF23 level in the group treated with ferric iron was half of that in the group treated with a conventional binder [135]. An RCT involving 441 dialysis patients showed a significant decrease in the phosphate level, significant increases in iron-related test parameters, and a significant reduction in the usage of iron and erythropoietin-simulating agent [136]. A meta-analysis confirmed that calcium levels were lower in patients who received ferric citrate compared with those who received other phosphate binders. In addition, ferric citrate had an additive effect on iron repletion and anemia control, which was predominantly associated with gastrointestinal side effects [137].

Give its nature as an iron agent, the risk of iron overload should be taken into account when administering ferric citrate. However, it is beneficial as iron replacement because it can be administered orally instead of intravenously. In addition, ferric citrate reduced FGF23 levels by more than half and also lowered the intact PTH (iPTH) level, thereby exerting effect on secondary hyperparathyroidism [138].

Meanwhile, sucroferric oxyhydroxide is a stable mixture of polynuclear iron (III)-oxyhydroxide, sucrose, and starch. After oral administration, the sucrose is broken down into glucose and fructose, and the starch into maltose and glucose, and finally the polynuclear iron (III)-oxyhydroxide is released. The phosphorous is bound via ligand exchange between a hydroxyl group and the hydrated water molecule of polynuclear iron (III)-oxyhydroxide and a phosphate ion. Iron, a biological element present in the body, is the key component of sucroferric oxyhydroxide. Phosphate binding occurs through ion exchange and iron ions are not expected to be released in the gastrointestinal tract. Nevertheless, it is not known whether the structure of sucroferric oxyhydroxide remains unaltered. If part of the structure is disturbed, some iron ions might be released and absorbed.

Sucroferric oxyhydroxide is indicated for use in dialysis patients. A phase 3 clinical study involving 644 patients compared the sucroferric oxyhydroxide treated group and the sevelamer treated group and showed significant decreases in the serum phosphate level in both groups, and a significantly smaller tablet number in the sucroferric oxyhydroxide treated group compared with the sevelamer treated group; the ferritin level was increased slightly, but transferrin saturation (TSAT), iron, and hemoglobin levels were stable [139]. A study (post hoc analysis of a randomized, 24-week Phase 3 study and its 28-week extension) that focused on iron-related parameters showed that changes in several parameters occurred within the first 24 weeks in both sucroferric oxyhydroxide- and sevelamer-treated groups, but to a lesser extent in longer-term observation. When sucroferric oxyhydroxide was compared with sevelamer, there were significantly greater increases in TSAT (+4.6% vs. +0.6%, *p* = 0.003) and hemoglobin levels (+1.6 g/L vs. −1.1 g/L during the first 24 weeks from baseline; the mean serum ferritin level was increased, although the difference between the groups was not significant (+119 ng/mL vs. +56.2 ng/mL) [140]. A different observational study also reported no significant changes in iron-related parameters in the 6-month observation period, indicating that, although both are iron-based, sucroferric oxyhydroxide appears to exert different effects compared with ferric citrate [141]. Also, in another study, 1059 patients were randomized 2:1 to sucroferric oxyhydroxide 1.0–3.0 g/day (n = 719) or sevelamer 2.4–14.4 g/day (n = 349) for 24 weeks. Sucroferric oxyhydroxide caused significant and sustained 30% reductions in serum phosphate level (*p* < 0.001) and significant 64% decreases in FGF23 level (*p* < 0.001). The iPTH level was decreased significantly at week 24 (*p* < 0.001) but returned to nearly the baseline level at week 52. Among bone resorption makers, tartrate-resistant acid phosphatase 5b decreased significantly (*p* < 0.001), whereas both bone formation markers, namely, bone-specific alkaline phosphatase and osteocalcin, increased [142].

After all, since both ferric citrate and sucroferric oxyhydroxide are based on iron. It can be said that there is a sense of security that iron is metal that has been used in the past, and there is less resistance to use with other heavy metals. However, the premise is different between the two that the iron is absorbed or not absorbed. Therefore, ferric citrate has medicinal property as an iron preparation, whereas sucroferric oxyhydroxide is not suitable for it. Ferric citrate can also be supplemented as iron, but conversely, decreasing or increasing the amount of ferric citrate only with respect to the phosphorus level may affect the decrease or otherwise overload of iron. On the other hand, sucroferric oxyhydroxide is not approved for use during the non-dialysis period in Japan, but it is not expected to play a role in iron supplementation, so on the contrary, iron may be absorbed in case, and along with the phosphorus level, iron dynamics need to be confirmed during administration.

## 10. Conclusions

The concept of CKD-MBD has been established, and several attempts have been made to address phosphate-associated toxicity and complications in CKD-MBD. Meanwhile, the αKlotho/FGF23 axis has been discovered and the phosphate metabolism mechanism, which involves the phosphate transporter Na/Pi, is becoming clear. Against this backdrop, dietary therapy and the use of a phosphate binders have become the basic approaches in managing phosphate levels in CKD patients. Unfortunately, both types of renal replacement therapy (hemodialysis and peritoneal dialysis) have limited phosphate excretion capacity, and so the phosphate balance tends to be positive even with a restricted diet. Despite the marked aging observed in CKD patients, extreme dietary restriction can cause both physical and mental functional impairments, and thus realistically there is a limit to protein restriction. In addition, phosphate intake is increasing due to the widespread use of phosphate-containing food additives. To address this situation, there is a movement to regulate the use of inorganic phosphate in foods, and development of new drugs that prevent phosphate absorption is under way. Phosphate absorption and excretion do not occur independently but are regulated by a network involving calcium, vitamin D, and PTH. Phosphate control requires continuous daily management, and further multidisciplinary studies are anticipated.

## Figures and Tables

**Figure 1 nutrients-13-01670-f001:**
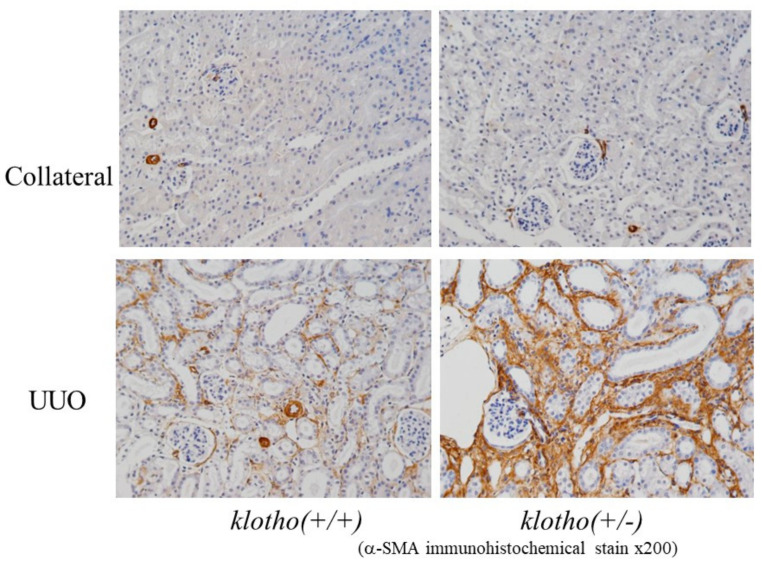
Representative sections immunohistochemically stained for α-SMA in the renal tubules in the reduced Klotho expression mice treated by unilateral ureteral obstruction. Renal fibrosis was induced by unilateral ureteral obstruction (UUO) in mice with reduced αKlotho expression α*kl* (*+/−*) mice and compared them with wild-type mice. The UUO kidneys from α*kl* (*+/−*) mice expressed significantly higher levels of fibrosis marker, α-smooth muscle actin (α-SMA), than those from wild-type α*kl* (*+/+*) mice. Adapted from reference [29].

**Figure 2 nutrients-13-01670-f002:**
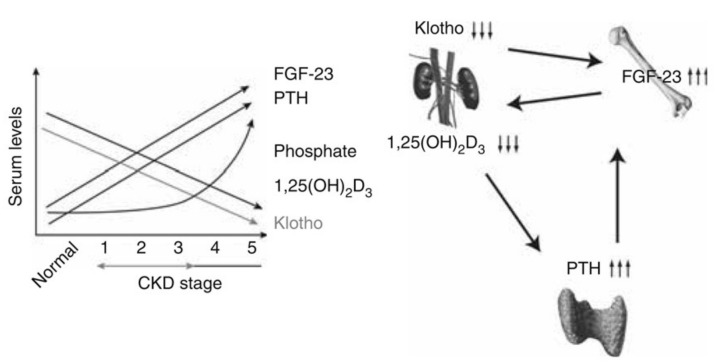
Changes in Klotho protein, FGF-23, PTH, 1,25(OH)2D3, and phosphate as CKD progresses. When Klotho expression first decreases, FGF-23 increases, lowering circulating 1,25(OH)2D3, which depresses Klotho expression further and increases PTH expression. Increased PTH induces further FGF-23 increases, causing large decreases in 1,25(OH)2D3 and large increases in PTH. This cycle results in hyperphosphatemia in late stages of CKD. CKD, chronic kidney disease; FGF-23, fibroblast growth factor-23; PTH, parathyroid hormone. Adapted from reference [38].

## Data Availability

Not applicable.

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
