# Peer review of "The Importance of Phosphate Control in Chronic Kidney Disease"

_nutrients, 2021, doi:10.3390/nu13051670_

Round 1

Reviewer 1 Report

The manuscript entitled " The importance of phosphate control in chronic kidney disease " by Tsuchiya  and Akihisa is a well structured review on the strategies to control phosphate levels in CKD-MB to improve prognosis.

Major points:

1.       The authors should refer Klotho as αKlotho in the whole manuscript, to distinguish it from its paralog βKlotho.

2.       The Abstract should clearly state the aim of this review.

3.       Edit lines 76 to 81 as it gives the impression that FGF23 is more important in regulating the action of sodium dependent phosphate transport than PTH.

4.       Edit lines 98 to 104  as αKlotho is not always necessary an obligatory co-receptor of FGF23 action, depends on the FGFR and the tissue.

5.       Lines 172-174: the reference is from 2014. It would be a nice part of the review to discuss the adequacy of the several ways to test αKlotho in blood samples.

Minor points:

1.       Line 53: change absorption by reabsorption

2.       Line 59-61: where is the missing percentage reabsorbed?

3.       Figure 2 left panel may have copy right issues with Kidney International as it is not an adaptation but a copy of the image. Check.

4.       Line 533: change FBF23 by FGF23

Reviewer 2 Report

see the attacmnent
